# mbkmeans: Fast clustering for single cell data using mini-batch *k*-means

**Stephanie C. Hicks**[1], **Ruoxi Liu**[2], **Yuwei Ni**[3¤], **Elizabeth Purdom**[4], **Davide Risso**[5]*

**1** Department of Biostatistics, Johns Hopkins Bloomberg School of Public Health, Baltimore, Maryland, USA,
**2** Department of Applied Mathematics and Statistics, Johns Hopkins University, Baltimore, Maryland, USA,
**3** Department of Healthcare Policy and Research, Weill Cornell Medical College, New York, New York, USA,
**4** Department of Statistics, University of California, Berkeley, Berkeley, California, USA, **5** Department of
Statistical Sciences, University of Padova, Padova, Italy

¤ Current address: R4 Technologies Inc., Ridgefield, Connecticut, USA
* risso.davide@gmail.com

**Data Availability Statement:** The data underlying the results presented in the study are publicly available as part of the TENxBrainData Bioconductor package available at: https://bioconductor.org/packages/TENxBrainData/.

## Abstract

Single-cell RNA-Sequencing (scRNA-seq) is the most widely used high-throughput technology to measure genome-wide gene expression at the single-cell level. One of the most common analyses of scRNA-seq data detects distinct subpopulations of cells through the use of unsupervised clustering algorithms. However, recent advances in scRNA-seq technologies result in current datasets ranging from thousands to millions of cells. Popular clustering algorithms, such as *k*-means, typically require the data to be loaded entirely into memory and therefore can be slow or impossible to run with large datasets. To address this problem, we developed the *mbkmeans* R/Bioconductor package, an open-source implementation of the mini-batch *k*-means algorithm. Our package allows for on-disk data representations, such as the common HDF5 file format widely used for single-cell data, that do not require all the data to be loaded into memory at one time. We demonstrate the performance of the *mbkmeans* package using large datasets, including one with 1.3 million cells. We also highlight and compare the computing performance of *mbkmeans* against the standard implementation of *k*-means and other popular single-cell clustering methods. Our software package is available in Bioconductor at https://bioconductor.org/packages/mbkmeans.

## Author summary

We developed the *mbkmeans* package (https://bioconductor.org/packages/mbkmeans) in Bioconductor, an open-source implementation of the mini-batch *k*-means algorithm. Our package allows for on-disk data representations, such as the common HDF5 file format widely used for single-cell data, that do not require all the data to be loaded into memory at one time.

**Funding:** This work has been supported by the National Institutes of Health grant R00HG009007 to SCH and by the the NIH BRAIN Initiative grant U19MH114830 (EP). This work was also supported by DAF2018-183201 (SCH, RL, YN, EP, DR) and CZF2019-002443 (SCH, RL, DR) from the Chan Zuckerberg Initiative DAF, an advised fund of Silicon Valley Community Foundation. EP was supported by a ENS-CFM Data Science Chair. DR was supported by Programma per Giovani Ricercatori Rita Levi Montalcini granted by the Italian Ministry of Education, University, and Research. The funders had no role in study design, data collection and analysis, decision to publish, or preparation of the manuscript.

This is a *PLOS Computational Biology* Software paper.

## Introduction

Unsupervised clustering algorithms are commonly used to divide a set of unlabeled observations into separate groups with similar traits [1, 2]. In particular, clustering algorithms are popular in single-cell transcriptomics, where datasets can consist of millions of unlabeled observations (or cells) [3, 4]. The goal in this setting is to group cells into distinct clusters with discrete labels that approximate true biological groups [5]. In this context, different clusters can be thought of as different cell types or cell states, which can be further explored in downstream analyses [4].

The most widely used partitional clustering algorithm is $k$-means [6–8]. The algorithm partitions $N$ cells into $k$ clusters each represented by a centroid, or mean profile, for the cells in the $k^{th}$ cluster. This algorithm is commonly used not only on its own, but also as a component of ensemble clustering [9, 10].

While $k$-means is easy to implement, it assumes that the user has enough computational resources (specifically RAM) to store the data and all intermediate calculations into memory. However, file sizes generated from scRNA-seq experiments can be on the order of tens to hundreds of gigabytes. For large enough data, $k$-means can be slow or completely fail if a user lacks sufficient computational resources. Ensemble clustering approaches that depend on the use of $k$-means [9, 10] run it multiple times (e.g., with different parameter values or on a different data subset) limiting the usability of these packages for large scRNA-seq datasets [11]. We note that our goal here is not to debate the relative merits of $k$-means as a clustering algorithm—$k$-means is a well-established method, which has been thoroughly investigated [12]—but to provide users with the ability to use the popular $k$-means algorithm on large single-cell datasets.

To address the problems of using $k$-means with large data, two solutions are (1) parallelization and (2) subsampling. Parallelization approaches typically leverage some combination of (i) *MapReduce* [13] concepts to handle a large volume of data over a distributed computing environment [14, 15], (ii) $k$-dimensional ($k$-d) trees to either optimize for the nearest centroid [16] or to partition datasets into subsets, representative of the larger dataset [17], and (iii) leverage multi-core processors [18]. While these approaches do improve the speed of $k$-means, they can be limited to the number of reducers for each centroid and can often require extensive computational resources. In contrast, subsampling approaches, such as the mini-batch $k$-means algorithm [19] work on small, random subsamples of data ("mini batches") that can fit into memory on standard computers. We would emphasize, however, that while mini-batch $k$-means only operates on small subsamples of the data at any one time, the algorithm still minimizes the same global objective function evaluated over all samples as in traditional implementations of $k$-means.

Current implementations of the mini-batch $k$-means algorithm [19] are available in standard programming languages such as in the *scikit-learn* machine learning Python library [20] or in the *ClusterR* R package [21]. However, these implementations either implicitly or explicitly require all the data to be read into memory, and therefore do not leverage the potential of the algorithm to provide a low memory footprint.

To address the described problems, we implemented the mini-batch $k$-means clustering algorithm in the open-source *mbkmeans* R package [22], providing fast, scalable, and memory-efficient clustering of scRNA-seq data in the Bioconductor framework [5, 23]. Like existing implementations, our package can be applied to in-memory data input for smaller datasets, but also to on-disk data, such as from the HDF5 file format [24], which is widely used

for distributing single-cell sequencing data. For on-disk input, *mbkmeans* leverages the sub-sampling structure of the algorithm to read into memory only the current "mini batch" of data at any given point, thereby greatly reducing the required memory (RAM) needed.

We evaluate the performance of *mbkmeans* compared to the standard *k*-means algorithm using the 1.3 million brain cells scRNA-seq data from 10X Genomics [25] and simulation studies. We demonstrate that our implementation constrains the memory usage and increases the speed of the clustering algorithm, performing orders of magnitude faster than the popular clustering algorithms Louvain [26] and Leiden [27], which are also frequently used for clustering single-cell sequencing data, without loss of accuracy as compared to the standard *k*-means algorithm. Our contribution is two-fold: we implement a mini-batch *k*-means algorithm for on-disk data, and we benchmark the performance of a non-trivial algorithm for HDF5 against its in-memory counterpart.

## Design and implementation

### Overview of *k*-means algorithm

Given a set of observations $\mathbf{Y} = \{\mathbf{y}_1, \mathbf{y}_2, \ldots, \mathbf{y}_N\}$ where each observation is a *G*-dimensional real vector, the optimization problem of *k*-means clustering is to partition the *N* observations into $k$ ($< N$) sets $\mathbf{S} = \{S_1, S_2, \ldots, S_k\}$ to minimize the within-cluster sum of squares (WCSS) or

$$\arg\min_{\mathbf{S}} \sum_{c=1}^{k} \sum_{\mathbf{y} \in S_c} ||\mathbf{y} - \mu_c||^2$$

where $\mu_c$ is the centroid of observations in $S_c$ and $||\cdot||$ denotes the L2 norm. Lloyd's algorithm [8] is the most widely used algorithm to solve this optimization, alternating between an **assignment step** and an **update step** until convergence.

### Overview of mini-batch *k*-means algorithm

Our mini-batch *k*-means implementation follows a similar iterative approach to Lloyd's algorithm. However, at each iteration *t*, a new random subset **M** of size *b* is used and this continues until convergence. If we define the number of centroids as *k* and the mini-batch size as *b* (what we refer to as the 'batch size'), then our implementation of mini-batch *k*-means follows that of *ClusterR* [21], and is briefly described here:

0. At $t = 0$: Initialize the set of *k* centroids $\hat{\mu}^{(0)} = (\hat{\mu}_1^{(0)}, \hat{\mu}_2^{(0)}, \ldots \hat{\mu}_k^{(0)})$.

1. For each $t \geq 1$: Randomly sample (without replacement) from **Y** a random subset **M** of size *b*. Update the estimates of the *k* centroids by performing the following two steps:

   (i). **Assignment**: Given the set of *k* centroids at $t-1$, or $\hat{\mu}^{(t-1)}$, compute the Euclidean distances between observations in **M** and the *k* cluster centroids. Assign each observation from **M** to the closest centroid to obtain a new set of observations per centroid $\mathbf{S}^{(t-1)} = \{S_1^{(t-1)}, \ldots, S_k^{(t-1)}\}$.

   (ii). **Update**: calculate the new centroids by averaging the coordinates of the observations from the mini-batch **M** assigned to each cluster to obtain $\hat{\mu}^{(t)} = (\hat{\mu}_1^{(t)}, \hat{\mu}_2^{(t)}, \ldots \hat{\mu}_k^{(t)})$. Steps (i-ii) are repeated until convergence using the difference in norm of the centroids, $||\hat{\mu}^{(t)} - \hat{\mu}^{(t-1)}||^2$.

2. Use the final estimates of the *k* centroids to assign all observations in **Y** to the cluster with the nearest centroid.

Like the $k$-means algorithm, the mini-batch $k$-means algorithm will result in different solutions at each run due to the random initialization point and the random samples taken at each point. Tang and Monteleoni [28] demonstrated that the mini-batch $k$-means algorithm converges to a local optimum. However, mini-batch $k$-means follows a different search path than $k$-means, and therefore does not necessarily converge to the same local optimum as the $k$-means algorithm.

## The *mbkmeans* software package

The *mbkmeans* software package implements the mini-batch $k$-means clustering algorithm described above and works with matrix-like objects as input. Specifically, the package works with standard R data formats that store the data in memory, such as the standard *matrix* class in base R and sparse and dense matrix classes from the *Matrix* R package [29], and with file-backed matrices, e.g., by using the HDF5 file format [24]. In addition, the package provides methods to interface with standard Bioconductor data containers such as the *SummarizedExperiment* [30] and *SingleCellExperiment* [31] classes.

We implemented the computationally most intensive steps of our algorithm in C++, leveraging the *Rcpp* [32] and *beachmat* [33] packages. Furthermore, we make use of Bioconductor's *DelayedArray* [34] framework, and in particular the *HDF5Array* [35] package to interface with HDF5 files. The *mbkmeans* package was built in a modular format that would allow it to easily operate on alternative on-disk data representations in the future. To initialize the $k$ centroids, the *mbkmeans* package uses the $k$-means++ initialization algorithm [36] with a random subset of $b$ observations (the batch size), by default. Finally, to predict final cluster labels, we use block processing through the *DelayedArray* [34] package to avoid working with all the data at once.

## Benchmarking datasets

To evaluate the performance of *mbkmeans*, we used: (i) simulated gene expression data from a mixture of $k$ Gaussian distributions (see S1 Text for details), and (ii) downsampled subsets of a real scRNA-seq dataset from 10X Genomics [25]. This scRNA-seq experiment was performed with the 10X Chromium Genomics platform [25] measuring the gene expression in mouse cells that came from three regions of the brain (cortex, hippocampus, and subventricular zone) and two mouse embryos (E18 C57BL/6 mice). After filtering out low-quality cells and lowly expressed genes, the dataset consists of $G = 11,720$ genes and $N = 1,232,055$ cells. We used all genes for the full analysis, while we focused on the 5,000 most variable genes for the subsampling analysis. We used the *TENxBrainData* Bioconductor data package [37] to access the data, which are stored as a dense matrix in a HDF5 file. See S1 Text for details about the processing of these data.

We evaluate both the memory usage and computing time using the `Rprof` and `proc.time` R functions. Both of these are reported from two independent computing systems: (i) an iMac with a 4.2GHz Intel Core i7-7700K CPU and 64 GB of RAM, which we refer to as "desktop" and (ii) a cluster node with 2.5GHz AMD Opteron Processor 6380 CPU, which we refer to as "HPC cluster" (high performance computing cluster).

## Results

One of the main purposes of unsupervised clustering algorithms for the analysis of scRNA-seq data is to empirically define groups of cells with similar expression profiles [5]. We explored the impact of the number of cells and batch sizes in a scRNA-seq dataset on the speed, memory-usage, and accuracy of *mbkmeans* as compared to $k$-means when predicting cluster labels.

Specifically, we evaluated the performance of (1) the standard (in-memory) $k$-means algorithm, as implemented in R [8], (2) mini-batch $k$-means applied to in-memory data, (3) and mini-batch $k$-means applied to an on-disk data representation (HDF5), where the last two implementations of mini-batch $k$-means are those available in our *mbkmeans* package. The results reported in the main text are based on our desktop configuration, but results for our HPC cluster configuration lead to similar conclusions, though all of the algorithms take slightly longer; the HPC cluster results are detailed in Supporting Information (S1, S2 and S11–S14 Figs).

### *mbkmeans* is fast and memory-efficient

Using downsampled datasets ranging from 75,000 to 1,000,000 observations from the 1.3 million mouse brain cells, we found that our on-disk (HDF5) *mbkmeans* uses dramatically less memory than either $k$-means or the in-memory *mbkmeans* for large scRNA-seq datasets (Fig 1A and S1 Table). There is almost no memory increase for datasets with larger sample sizes using our on-disk implementation of *mbkmeans*, and we can cluster 1 million cells with only 1.55GB of RAM, as compared to 39.4 GB for the in-memory version. We were not able to compare to standard $k$-means at large sample sizes due to lack of sufficient memory; however, when using 300,000 cells, $k$-means used 52 GB as opposed to 0.98GB with the on-disk *mbkmeans*. In addition, the in-memory *mbkmeans* uses far less memory than $k$-means, requiring only 11.95GB for 300,000 cells. In addition, we compared our in-memory *mbkmeans* implementation to the mini-batch $k$-means algorithm implemented in the *ClusterR* [21] R package (S1 Fig). We found *mbkmeans* was more memory-efficient compared to *ClusterR*, demonstrating the improvements over $k$-means are not only achieved from the mini-batch $k$-means algorithm itself, but also in the implementation of our *mbkmeans* software package.

Furthermore, we found both of our *mbkmeans* implementations are significantly faster compared to $k$-means (Fig 1B and S1 Table). Specifically, we can cluster 1 million cells in 9.8 and 7.8 minutes (mean values across 10 runs) for in-memory and on-disk implementations of mini-batch $k$-means, respectively, compared to 36.6 minutes using in-memory $k$-means for 300,000 cells ($k$-means fails to complete with larger datasets). In addition, we found *mbkmeans* performed similarly in computational time compared to *ClusterR*, leading to no loss in performance of speed (S2 Fig).

### *mbkmeans* is accurate

Using simulated gene expression data, we explored the effect of increasing sizes of datasets, as well as batch sizes, on the performance of the algorithms. We found that using datasets with a batch size $b = 500$ observations or larger led to no loss in accuracy with respect to $k$-means, based on ARI (Fig 2A and S2 Table) and WCSS (Fig 2B and S2 Table). In addition, we considered a wider range of sizes of datasets and found consistent ARI and WCSS results (S3–S6 Figs). This confirms the results of [19].

Next, we assessed the WCSS using the 1.3 million mouse brain cells by downsampling to similar dataset sizes ($N = 5,000$, 10,000, and 25,000). We found similar results to simulated data (Fig 2C and S2 Table) with no substantial difference in the minimum observed WCSS between the $k$-means and *mbkmeans* algorithms (using $k = 15$) if using a batch size of $b = 500$ observations or more. These results demonstrate that, as long as the batch size of the *mbkmeans* is not unreasonably small (at least $500 - 1, 000$ cells in a batch), the *mbkmeans* algorithm is as accurate as the standard $k$-means algorithm. We note that these results do not depend on our implementation, but on the mini-batch $k$-means algorithm, as both *mbkmeans*

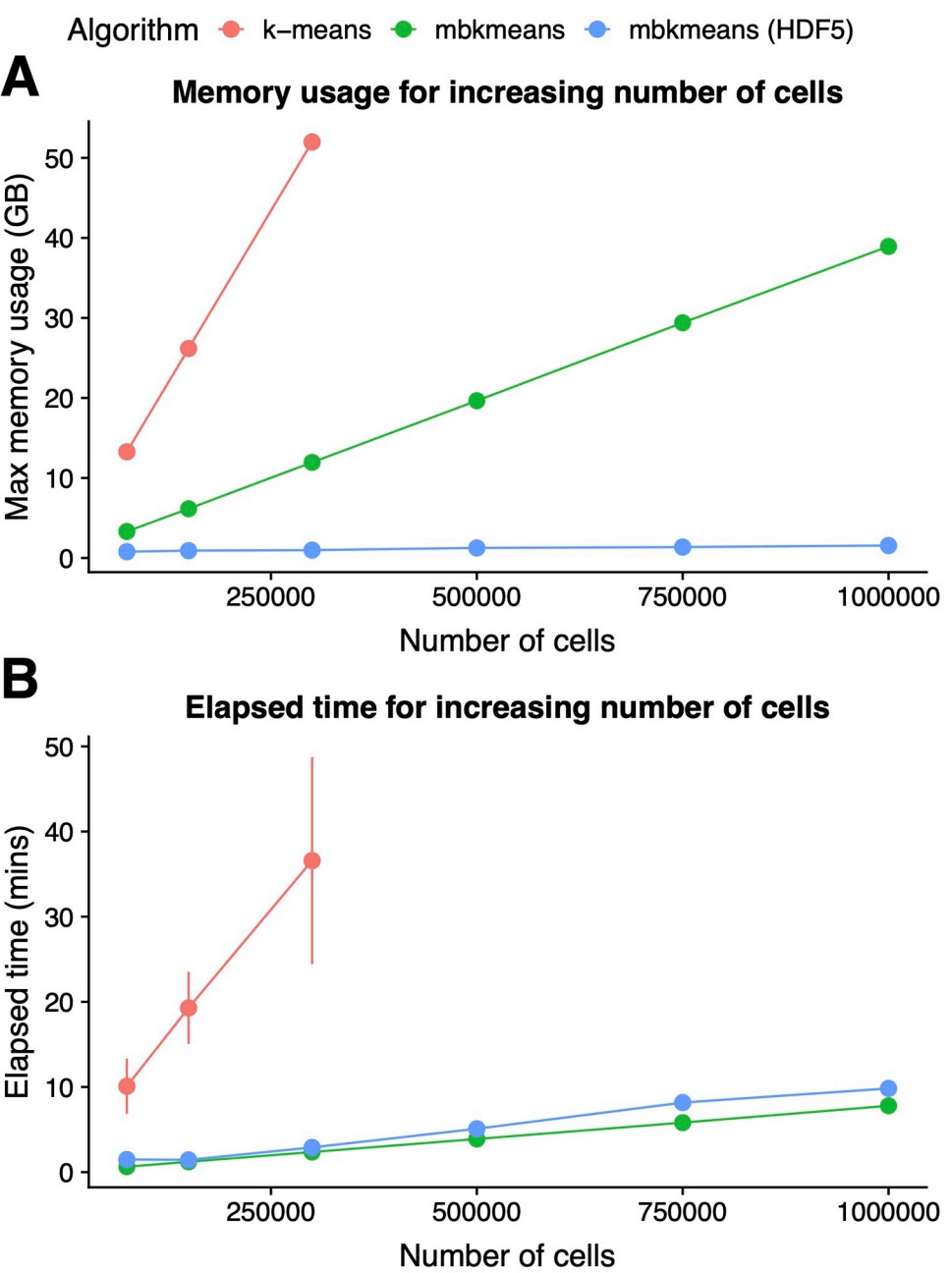

**Fig 1. *mbkmeans* uses less memory and is faster than *k*-means.** Performance evaluation (y-axis) of **(A)** maximum memory (RAM) used (GB) and **(B)** elapsed time (minutes) (repeated 10 times) for increasing sizes of datasets (x-axis) with $N$ = 75,000, 150,000, 300,000, 500,000, 750,000, and 1,000,000 observations and $G$ = 5,000 genes, using our desktop configuration. Results for *mbkmeans* are in green (in-memory) and blue (on-disk); *k*-means is in red. We used $k$ = 15 for both algorithms and used a batch size of $b$ = 500 observations for *mbkmeans*.

and the implementation in the *ClusterR* [21] R package lead to comparable WCSS values (S7 Fig).

Finally, because one of the user-defined parameters in the *k*-means and mini-batch *k*-means algorithms is the number of centroids (*k*), we investigated the impact of *k* on both memory-usage and accuracy. We found that the maximum memory (GB) used was

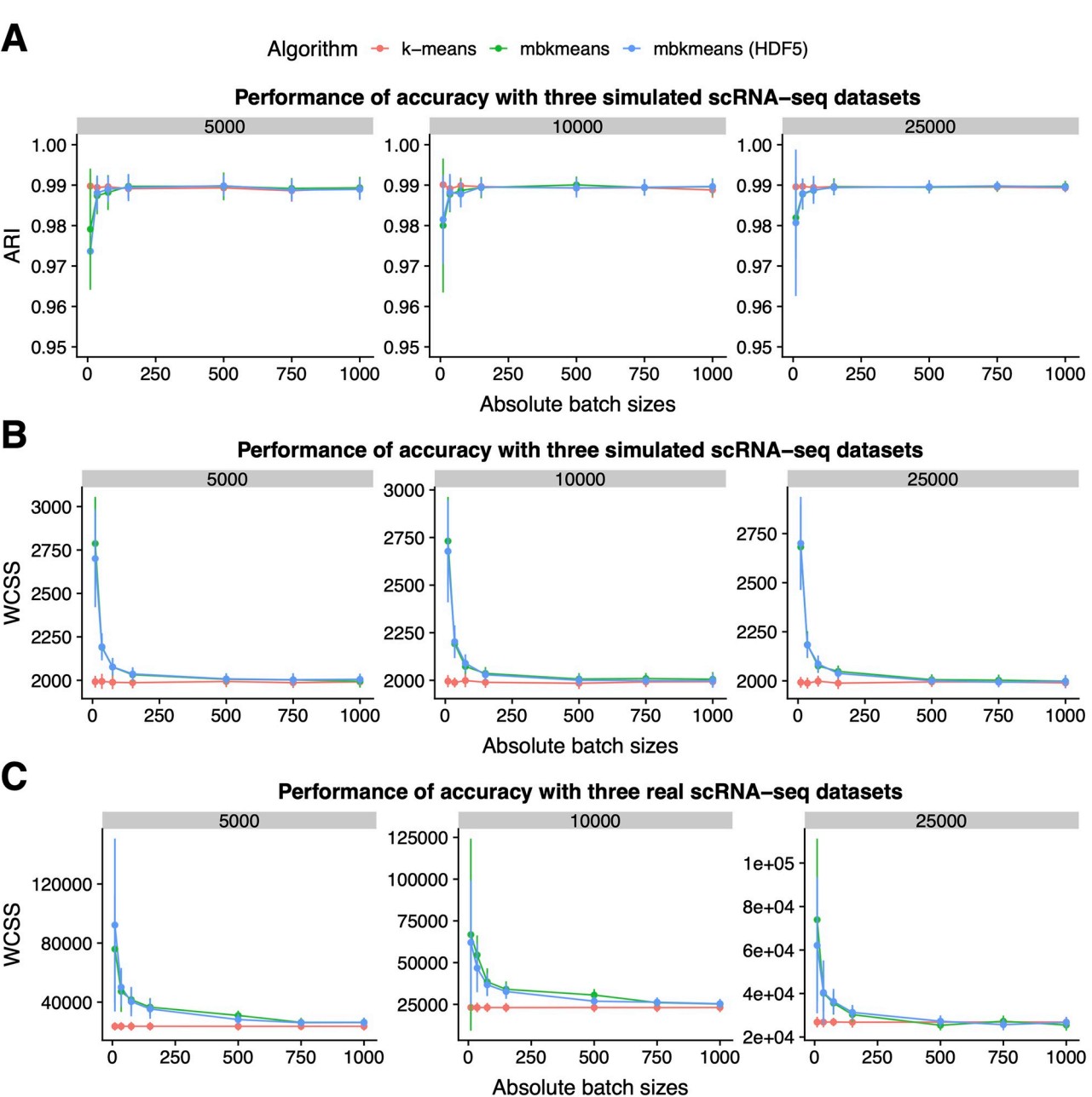

**Fig 2. The accuracy of *mbkmeans* depends on batch size.** Performance evaluation (y-axis) with **(A)** adjusted Rand index (ARI) and **(B)** within clusters sum of squares (WCSS) for increasing batch sizes ranging from 75 to 1000 cells (x-axis) using simulated gene expression data ($G$ = 1000) with a fixed $k$ = 3 true centroids with three sizes of datasets ($N$ = 5000, 10000, 25000). **(C)** WCSS (y-axis) for increasing batch sizes (x-axis) using real scRNA-seq gene expression data from 10X Genomics and $k$ = 15 for both algorithms. ARI and WCSS is reported as an average across 50 runs.

not affected by $k$ either in the in-memory or on-disk implementations for small ($N$ = 25,000) or large ($N$ = 1,000,000) datasets (S8 Fig). However, using simulated scRNA-seq data we found that accuracy—using ARI (S9 Fig) or using WCSS (S10 Fig)—varied as a function of $k$ with the highest accuracy resulting when $k$ = 15 (the true simulated centroids), as expected.

### Impact of *mbkmeans* batch size on speed and memory-usage

The choice of batch size (*b*) in *mbkmeans* can in principle have an impact on how long it takes the algorithm to run and how much memory is used. We used the downsampled 1.3 million mouse brain cells dataset with $N = 1,000,000$ observations and considered the effect of increasing batch sizes for both the in-memory and on-disk implementations of *mbkmeans*. We found that there was no impact on maximum memory usage (Fig 3A and S3 Table) nor speed (Fig 3B and S3 Table) for batch sizes up to $b = 10,000$. This is perhaps unsurprising, since computations in memory on datasets of size 10,000 are routine and unlikely to have noticeable differences in time and memory. For larger batch sizes, we started to see noticeable increases in demand for memory and larger time to run *mbkmeans*.

All considered, we recommend users to use a batch size of $b = 10,000$, which balances both the results from accuracy (Fig 2), memory-usage and speed (Fig 3). Furthermore, since the initialization uses by default the same number of cells as batch size, this gives a robust sample for determining the initial start point.

Combined with the results of accuracy, these results demonstrated that the algorithm is not sensitive to the batch size parameter, with values in the range of $b = 500$ to 10,000 giving comparable results with respect to both accuracy and computational performance.

### HDF5 file format geometry can improve performance

How the data are stored (or 'chunked') and accessed in the HDF5 format (or different file geometries) can affect the speed of accessing the data. For example, the indexing can be based on columns or rows of the data (vertical or horizontal slices), or other sub-matrices (rectangles). In the case of a matrix (two-dimensional array), the default chunk size in the *HDF5Array* package selects a rectangle that satisfies the following constraints: (i) its area is less than 1,000,000; (ii) it fits in the original matrix; (iii) its shape is as close as possible as the shape of the original matrix (the two dimensions are in the same proportion).

We investigated the choice of different file chunking geometries on the performance of the *mbkmeans* algorithm, and we found that the best choice for minimizing the maximum RAM used is to index a HDF5 file by the cells (or observations) of the matrix (Fig 4A and S4 Table); indexing by cells also improves the speed of the algorithm (Fig 4B and S4 Table).

In contrast, if a file is indexed by genes, we found the on-disk *mbkmeans* implementation required twice as much RAM—though even then it is still a relatively small memory footprint compared to an in-memory version. We also considered indexing a HDF5 file by the entire matrix in one chunk ("single chunk" in Fig 4), which is the default in the HDF5 library format [24]. In this case, we found that even for a small number of cells, there is an substantial memory cost for this naive geometry.

Finally, we note that while the default geometry implemented in *HDF5Array* is not optimal for clustering, there are many different components in a standard scRNA-seq pipeline, with clustering typically not being the slowest step (see 'A complete analysis of a large single-cell dataset' Section below), and thus the choice of geometry may ultimately be better determined by the performance of these other steps.

### A complete analysis of a large single-cell dataset

We analyzed the full 1.3 million mouse brain cells with all the steps in a standard scRNA-seq analysis [5]: quality control and filtering, normalization, dimensionality reduction using Principal Components Analysis (PCA), and finally clustering for detection of subtypes. For all of the non-clustering steps, we used recent packages in Bioconductor that operate directly on HDF5 files. Specifically, we used *scater* [38] for quality control and dimensionality

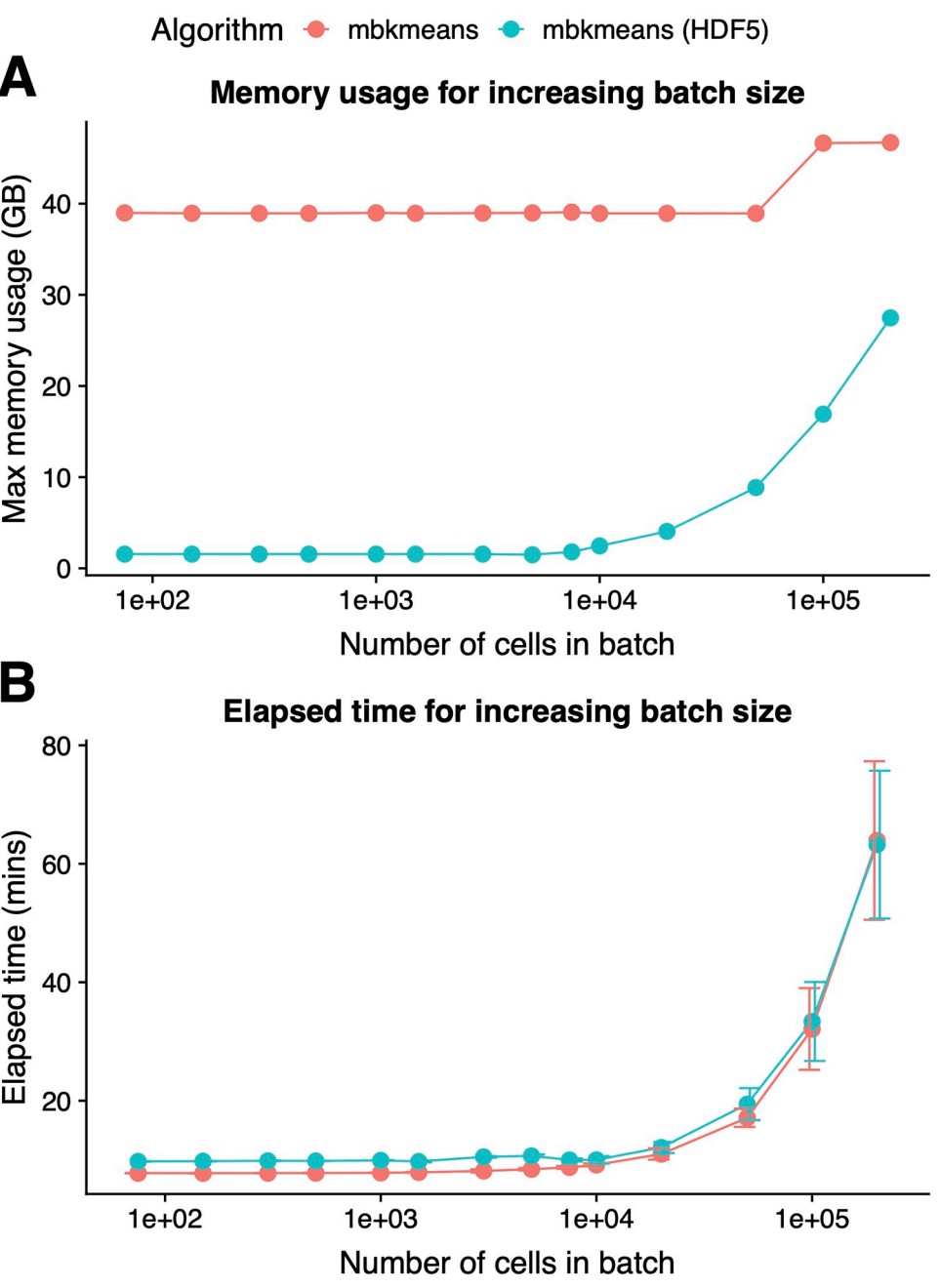

**Fig 3. The speed and memory-usage of *mbkmeans* depends on batch size.** Performance evaluation (y-axis) of **(A)** maximum memory (RAM) used (GB) and **(B)** elapsed time (minutes) for increasing batch sizes (x-axis) with $b = 75$, 150, 300, 500, 1,000, 1,500, 3,000, 5,000, 7,500, 10,000, 20,000, 50,000, 100,000, and 200,000 with a dataset of size $N = 1,000,000$ observations using our desktop configuration. Results for *mbkmeans* in-memory are in red and and on-disk in blue. We used $k = 15$ for the number of centroids.

reduction, via the *BiocSingular* package, which implements the implicitly restarted Lanczos bidiagonalization algorithm (IRLBA) [39], and *scran* [40] for normalization. We note that we used *mbkmeans* as a preliminary step to create homogeneous cell groups for *scran* normalization (see [40] for details). Our *mbkmeans* package complements these existing packages,

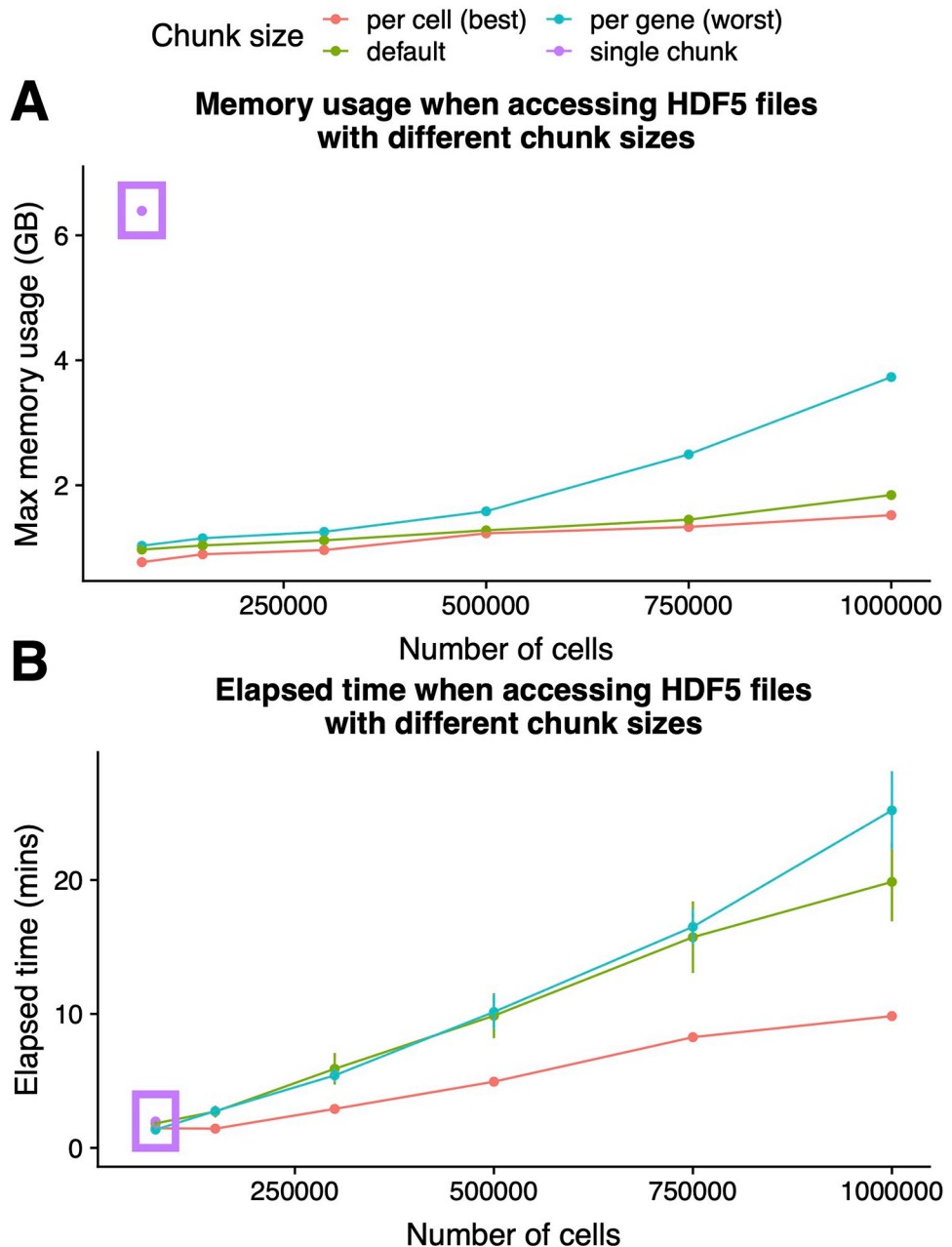

**Fig 4. The speed and memory-usage of the on-disk *mbkmeans* implementation depends on the structure of the on-disk file.** Performance evaluation (y-axis) of **(A)** maximum memory (RAM) used (GB) and **(B)** elapsed time (minutes) (repeated 10 times) for increasing sizes of datasets (x-axis) with $N$ = 75,000, 150,000, 300,000, 500,000, 750,000, and 1,000,000 observations using our desktop configuration. Results for indexing a HDF5 file by gene is blue, by cell is red, as a single chunk is purple and the default indexing is green. The single chunk was only able to run for the smallest dataset size ($N$ = 75,000). We used $k$ = 15 and used a batch size of $b$ = 500 observations for *mbkmeans*.

allowing us to demonstrate a complete HDF5-ready pipeline for scRNA-seq analysis in R/Bioconductor.

We note that because clustering is generally performed after a dimensionality reduction (e.g. reducing to the top 50 principal components or PCs), the size of the data matrix to be clustered is significantly reduced in complexity and can be potentially performed in-memory,

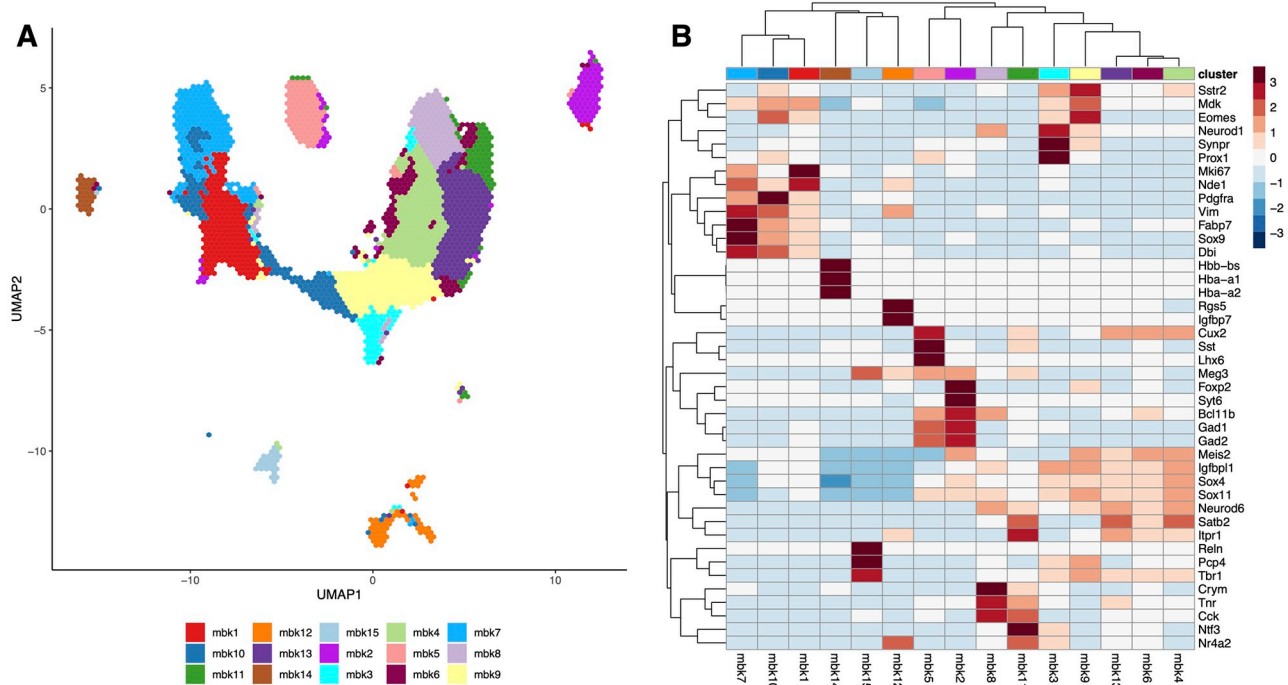

**Fig 5. Results of full analysis on 1.3 million mouse brain cells. (A)** Hexbin plot [54] of the UMAP representation of the 1.3 million cells, color coded by the clusters found via *mbkmeans*. **(B)** Heatmap of the average gene expression of each of the 15 clusters found by *mbkmeans* for 42 marker genes.

even with 1.3 million cells. However, some normalization methods, such as *scran*, make use of an initial clustering to improve the accuracy of the normalization, which is before the dimensionality reduction step; in this case the dimension of the clustering problem is over all cells and all expressed genes and benefits greatly from using on-disk clustering.

In S5 Table, we show the compute time for each step of the pipeline. The clustering of the data with *mbkmeans*, even with all genes, was a very quick part of the pipeline, taking roughly 9 minutes (with $k = 15$ and batch size of 500); in contrast, normalization took 5 hours, and IRLBA PCA took 96 hours. On the computationally simpler problem of clustering on only the top 50 PC dimensions, which we performed via the in-memory *mbkmeans*, the clustering took only about 30 seconds—quickly enough to rerun the clustering algorithm with different number of clusters or on random subsamples of the data for stability analysis.

We compared our pipeline with two alternative approaches, based on Louvain and Leiden clustering, implemented in Bioconductor and in the *scanpy* Python package [41], respectively. Specifically, after obtaining the top 50 PC dimensions, we computed the shared nearest neighbor (SNN) graph and we detected communities in the network via the Louvain algorithm [26]. Alternatively, after normalization and PCA, we performed batch correction using the BBKNN method [42] and Leiden clustering [27] on the resulting network.

Comparing the resulting partitions, we observed that the clustering results of the three methods are qualitatively similar (S15 and S16 Figs) and that both Louvain and Leiden are considerably slower than *mbkmeans*. The Louvain algorithm took 35.5–156 minutes (depending on whether finding an approximate or exact solution) and the Leiden algorithm took 48.5 minutes (S7 Table); this is in contrast to 30 seconds with *mbkmeans*.

In Fig 5A we show the UMAP of the full dataset, with cells color-coded by their assignment into the 15 final clusters found by applying *mbkmeans* on the first 50 PCs. We note that the

*mbkmeans* clustering is not driven by batch effects, as shown by the Adjusted Rand Index (ARI) of the cluster labels and the "mouse" variable (ARI = 0.04). *mbkmeans* clearly identified distinct outlying groups of cells, such as clusters 2, 5, 12, 14, and 15 (S6 Table). The remaining clusters are not easily separated by eye in two dimensions, making it difficult to discern whether *mbkmeans* is identifying meaningful clusters. To further evaluate the clusters, we accumulated a list of genes previously shown to discriminate known subtypes of cells in developing and adult mouse brains [43–49]. We show the average expression of each cluster for these marker genes in Fig 5B. Many of the clusters identified by *mbkmeans* have unique expression of these marker genes, indicating that *mbkmeans* is finding meaningful biological clusters. For example, clusters 1, 7, and 10—the boundaries of which are not obviously distinguished by eye on the UMAP representation—all correspond to Radial Precursors [47], but they each have clear markers that distinguish them, suggesting that they correspond to different developmental stages (see S6 Table). Similarly, cluster 8 (expressing the L2/3 marker *Crym* [44]) and 11 (expressing the L5/6 marker *Ntf3* [43]) represent two distinct pyramidal excitatory neuron populations, possibly residing in different cortical layers.

## Availability and future directions

A major challenge in the analysis of scRNA-seq data is the scalability of analysis methods as datasets increase in size over time. This is particularly problematic as experiments now frequently produce millions of cells [50–53], possibly across multiple batches, making it challenging to even load the data into memory and perform downstream analyses including quality control, batch correction and dimensionality reduction. Providing analysis methods, such as unsupervised clustering, that do not require data to be loaded into memory is an imperative step for scalable analyses. While large-scale scRNA-seq data are now routinely stored in on-disk data formats (e.g. HDF5 files), the methods to process and analyze these data are lagging.

To address this, we have developed an open-source implementation of the mini-batch *k*-means algorithm to provide an unsupervised clustering algorithm scalable to millions of observations. Unlike other existing implementations of mini-batch *k*-means, our algorithm harnesses the structure of the mini-batch *k*-means algorithm to only read in the data needed for each batch, controlling memory usage for large datasets. This makes our implementation truly scalable and applicable to both standard in-memory matrix objects, including sparse matrix representations, and on-disk data representations that do not require all the data to be loaded into memory at any one time, such as HDF5 matrices. We have demonstrated the performance improvement of the *mbkmeans* package across a range of different sized datasets, both with simulated and real single-cell datasets. We have also benchmarked an end-to-end Bioconductor pipeline, which includes using *mbkmeans* for subtype discovery, on a 1.3 million scRNA-seq dataset.

Our implementation of mini-batch *k*-means is available as the open-source *mbkmeans* package in Bioconductor (https://bioconductor.org/packages/mbkmeans). The analyses in this manuscript were performed using *mbkmeans* version 1.4.0, and all code to replicate the analyses is available at: https://github.com/stephaniehicks/benchmark-hdf5-clustering.

## Supporting information

**S1 Fig. Memory-usage reported for both desktop and HPC cluster configurations corresponding to Fig 1.**
(PDF)

**S2 Fig. Elapsed time (minutes) reported for both desktop and HPC cluster configurations corresponding to Fig 1.**
(PDF)

**S3 Fig. Accuracy (ARI) corresponding to Fig 2A using simulated data ($N$ = 500, 2,000, 4,000, 5,000, 6,000, 8,000, 10,000, 25,000 observations) reported using our desktop configuration.**
(PDF)

**S4 Fig. Accuracy (WCSS) corresponding to Fig 2B using simulated data ($N$ = 500, 2,000, 4,000, 5,000, 6,000, 8,000, 10,000, 25,000 observations) reported using our desktop configuration.**
(PDF)

**S5 Fig. Accuracy (ARI) corresponding to Fig 2A using simulated data ($N$ = 500, 2,000, 4,000, 5,000, 6,000, 8,000, 10,000, 25,000 observations) reported using a HPC cluster.**
(PDF)

**S6 Fig. Accuracy (WCSS) corresponding to Fig 2B using simulated data ($N$ = 500, 2,000, 4,000, 5,000, 6,000, 8,000, 10,000, 25,000 observations) reported using a HPC cluster.**
(PDF)

**S7 Fig. Accuracy (WCSS) corresponding to Fig 2C using real scRNA-seq gene expression data from 10X Genomics ($N$ = 5,000, 10,000, and 25,000 observations) and $k$ = 15 for all algorithms. WCSS is reported as an average across 50 runs for both a desktop and HPC cluster.**
(PDF)

**S8 Fig. Memory usage with two sizes of simulated scRNA-seq datasets and three absolute batch sizes (75, 500, 1000) with the true number of clusters as k = 15.**
(PDF)

**S9 Fig. Performance evaluation of ARI with increasing estimated cluster centroids $k$ using *mbkmeans* and $k$-means using our desktop and HPC cluster configurations.** We simulated gene expression data with 15 true centroids for two sizes of datasets ($N$ = 25000, 100000, both using $G$ = 1000 genes) considered three absolute batch sizes of cells ($b$ = 75, 500, 1000) for *mbkmeans* (both in memory and on-disk using HDF5 files using our desktop configuration). We show the impact of increasing the number of estimated cluster centroids $k$ used in the clustering algorithm (x-axis) on the adjusted Rand index (ARI) performance metric (y-axis).
(PDF)

**S10 Fig. Performance evaluation of WCSS with increasing estimated cluster centroids $k$ using *mbkmeans* and $k$-means using our desktop and HPC cluster configurations.** We simulated gene expression data with 15 true centroids for two sizes of datasets ($N$ = 25000, 100000, both using $G$ = 1000 genes) considered three absolute batch sizes of cells ($b$ = 75, 500, 1000) for *mbkmeans* (both in memory and on-disk using HDF5 files using our desktop configuration). We show the impact of increasing the number of estimated cluster centroids $k$ used in the clustering algorithm (x-axis) on the within clusters sum of squares (WCSS) performance metric (y-axis).
(PDF)

**S11 Fig. Memory-usage reported for both desktop and HPC cluster configurations, corresponding to Fig 3.**
(PDF)

**S12 Fig. Elapsed time (minutes) reported for both desktop and HPC cluster configurations, corresponding to Fig 3.**
(PDF)

**S13 Fig. Memory-usage reported for both desktop and HPC cluster configurations, corresponding to Fig 4.**
(PDF)

**S14 Fig. Elapsed time (minutes) reported for both desktop and HPC cluster configurations, corresponding to Fig 4.**
(PDF)

**S15 Fig. Comparison between *mbkmeans* ($k$ = 15) and Louvain clustering in the space of the top 50 Principal Components of the 1.3 million mouse brain cells dataset.** See S1 Text for details on the Louvain clustering.
(PNG)

**S16 Fig. Comparison between *mbkmeans* ($k$ = 15) and Leiden clustering in the space of the top 50 Principal Components of the 1.3 million mouse brain cells dataset.** See S1 Text for details on the Leiden clustering.
(PNG)

**S1 Table. Performance evaluation for memory-usage and elapsed time as reported in Fig 1.**
We report the maximum memory (RAM) used (GB) and averaged elapsed time (minutes) for increasing sizes of datasets with $N$ = 75,000, 150,000, 300,000, 500,000, 750,000, and 1,000,000 observations and 5,000 genes using our desktop computer configuration. The average elapsed time (elapsed_mean) and standard deviation (elapsed_sd) of ten runs is reported in the table. We used $k$ = 15 for both algorithms and used a batch size of $b$ = 500 observations for *mbkmeans*.
(PDF)

**S2 Table. Performance evaluation for accuracy as reported in Fig 2.** We report the adjusted Rand index (ARI) and within-cluster sum of squares (WCSS) averaged across 50 replicates for increasing sizes of datasets with $N$ = 5,000, 10,000, and 25,000 observations and increasing batch sizes $b$ = 10, 35, 75, 150, 500, 750, 1,000 using our desktop computer configuration. The average ARI for simulated data (ari_sim_mean) and standard deviation (ari_sim_sd), average WCSS for simulated data (wcss_sim_mean) and standard deviation (wcss_sim_sd), average WCSS for real scRNA-seq data (ari_real_mean) and standard deviation (ari_real_sd), is reported in the table. We used $k$ = 3 for simulated data and $k$ = 15 for real scRNA-seq data for all algorithms.
(PDF)

**S3 Table. Performance evaluation for memory-usage and elapsed time reported in Fig 3.**
We report the maximum memory (RAM) used (GB) and averaged elapsed time (minutes) for increasing batch sizes with $b$ = 75, 150, 300, 500, 1,000, 1,500, 3,000, 5,000, 7,500, 10,000, 20,000, 50,000, 100,000, 200,000 with a dataset of size $N$ = 1,000,000 observations and 5,000 genes using our desktop computer configuration. The average (elapsed_mean) and standard deviation (elapsed_sd) of ten runs is reported in the table. We used $k$ = 15 for the number of centroids in *mbkmeans*.
(PDF)

**S4 Table. Performance evaluation for memory-usage and elapsed time as reported in Fig 4.** We report the maximum memory (RAM) used (GB) and averaged elapsed time (minutes) for increasing sizes of datasets with $N = 75,000$, $150,000$, $300,000$, $500,000$, $750,000$, and $1,000,000$ observations and $5,000$ genes using our desktop computer configuration. The average elapsed time (elapsed_mean) and standard deviation (elapsed_sd) of ten runs is reported in the table. The single chunk was only able to run for the smallest dataset size ($N = 75,000$). We used $k = 15$ and used a batch size of $b = 500$ observations.
(PDF)

**S5 Table. Computational time for each of the steps of the pipeline for the full 1.3 million mouse brain cells.**
(PDF)

**S6 Table. Identification of *mbkmeans* clusters with Marker Genes.**
(PDF)

**S7 Table. Computational time for graph-based clustering.** SNN: Shared Nearest Neighbors; BBKNN: Batch-Balanced K-Nearest Neighbors.
(PDF)

**S1 File. Current implementation of the *mbkmeans* package as a tar file.**
(GZ)

**S1 Text. Supplementary text.**
(PDF)

## Acknowledgments

The authors would like to thank Hervé Pagès, Mike Smith, Kasper Hansen, and Pete Hickey for helpful discussions on using HDF5 files in the Bioconductor framework, and Lampros Mouselimis (author of the *ClusterR* package) for exposing some internal functions for use in *mbkmeans*.

## Author Contributions

**Conceptualization:** Stephanie C. Hicks, Elizabeth Purdom, Davide Risso.

**Formal analysis:** Stephanie C. Hicks, Ruoxi Liu, Davide Risso.

**Funding acquisition:** Stephanie C. Hicks, Davide Risso.

**Methodology:** Stephanie C. Hicks, Elizabeth Purdom, Davide Risso.

**Project administration:** Davide Risso.

**Software:** Yuwei Ni, Davide Risso.

**Supervision:** Stephanie C. Hicks, Davide Risso.

**Visualization:** Stephanie C. Hicks, Ruoxi Liu, Davide Risso.

**Writing – original draft:** Stephanie C. Hicks, Elizabeth Purdom, Davide Risso.

**Writing – review & editing:** Stephanie C. Hicks, Ruoxi Liu, Yuwei Ni, Elizabeth Purdom, Davide Risso.

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
