## [Decision Letter · Decision Letter 0]

17 Sep 2020

Dear Dr. Risso,

Thank you very much for submitting your manuscript "mbkmeans: fast clustering for single cell data using mini-batch k-means" for consideration at PLOS Computational Biology. As with all papers reviewed by the journal, your manuscript was reviewed by members of the editorial board and by several independent reviewers. The reviewers appreciated the attention to an important topic. Based on the reviews, we are likely to accept this manuscript for publication, providing that you modify the manuscript according to the review recommendations.

Specifically, please revise the manuscript to include information about how mbkmeans compares to other clustering tools, what is the major novelty and advantages.

Sincerely,

Dina Schneidman

Software Editor

PLOS Computational Biology

[LINK]

Reviewer's Responses to Questions

**Comments to the Authors:**

Reviewer #1: Hicks et al present "mbkmeans: fast clustering for single cell data using mini-batch k-means". The paper is very well written and looks very logical. mbkmeans development is very timely, since the size of scRNAseq data has grown dramatically in the last few years. mbkmeans solves the problem of in memory clustering by operating with data on disk due to utilisation of hdf5 format. The package is already available on bioconductor and is well integrated with the bioconductor universe. The datasets used are of appropriate size and quality. I recommend this paper for publication.

Reviewer #2: mbkmeans (multi-batch K-means) is a package that fills a gap in the R/Bioconductor ecosystem: tractable data integration of total mRNA copy number estimates from millions of cells, by k-means clustering. As such, alternatives that exist for comparison appear to require stepping out of the controlled, vetted ecosystem into something like scanpy or bbknn. The latter, batch-balanced K-nearest neighbors (Polanski et al., Bioinformatics 2020), is perhaps an obvious target of comparison for mbkmeans (or perhaps not -- mbkmeans does not seem to make explicit claims about the data integration utility of its fast and memory-efficient implementation, which may or may not be relevant). Polanski et al. did not appear in the References section of this paper, and perhaps the authors view it as out of scope, but while scaling up clustering to millions of cells is useful, if the clustering ends up reinforcing artifacts in the provided data, this would seem to reduce the utility. Most users operating at the scale of millions of cells are likely to be integrating datasets, either before clustering, or as part of it, so I feel like this is an issue that needs to be discussed, if not necessarily dealt with.

The HDF5 clustering benchmark code is useful as a standalone harness, and the authors are to be commended for including it.

In running the software from its vignette, I found that the necessary library() calls to mbkmeans, TENxPBMCData, scater, and DelayedMatrixStats were omitted from the code snippets. Thus, a user attempting to copy and paste directly from the package vignette will encounter errors. This should be fixed; the package itself is a part of any application note, as one cannot apply a paper directly to a computational problem, and the vignette for a package is an integral part of both its documentation and (to some extent) user testing. Once these missing calls were included, the vignette runs without incident, although the demonstration sample size is so small as to somewhat diminish the utility of the vignette in playing with parameters. From the point of view of a user, the package vignette is vastly more likely to serve as the point of entry than a paper whose performance evaluations are a bit peripheral to the question of "should I use this method at all, or just use scanpy+bbknn+UMAP+leiden". The latter has some sleight of hand involved (downsampling).

In the above vignette, there does not appear interpretation of why different clustering results are obtained from mini-batch K-means and full K-means (as demonstrated in vignette steps 3.1.2 and 4), nor with k-means++ initialization, especially given the notes in the paper itself. In the paper we find a note that, for batches of k=500 or larger, minibatch k-means recovers the results of classical K-means ("mkbmeans is accurate"). The authors note that the employed minibatch k-means method from ClusterR is essentially scaled up and adapted to an HDF5 backend, which (while more challenging than it may appear) is not inherently of broad interest. Thus the demonstration that the package does in fact recover results previously feasible only with much more RAM and CPU seems important.

Similarly, hyperparameter tuning is an issue in almost all single-cell RNAseq tasks, especially when it is a primary free parameter (number of centroids) in an algorithm. This seems to deserve somewhat more discussion or automation; the use of the adjusted Rand index (ARI) to gauge fit is useful, so a method to automate the process of evaluation seems germane to the package. Alternatively, implementations of parallel Bayesian optimization exist (e.g. https://cran.r-project.org/web/packages/ParBayesianOptimization/vignettes/tuningHyperparameters.html ) and might be appropriate for some of these subtasks. The authors note that existing clustering methods are memory-intensive and implicitly require loading of entire matrices into memory, whereas mbkmeans does not. The issue of downsampling (property-preserving or otherwise) as a workaround for some of these issues is not really touched upon other than to note that mbkmeans "samples" only the minibatch-sized block of observations required to iteratively employ the method. Whether structure beyond 5-15 centroids is effectively recovered in atlas-sized datasets of the sort that primarily motivate this memory-efficient work is not really discussed in the manuscript that I could see.

I am less interested in "comparing" mbkmeans against itself on a 1.3M-cell dataset than against something like bbknn and/or leiden downstream of scanpy, and I feel that the authors are remiss in not doing so, even if only to contrast the two in terms of platform. The supplementary figures for the bbknn paper are remarkably effective in demonstrating a use case where the popular Harmony method fails to recover biologically relevant similarities (in their use case, the single best studied somatic stem cells in all of biology). Something along these lines seems appropriate in an Application Note, to motivate _why_ a user would apply mbkmeans to their data, instead of simply dumping it to an .h5ad file and ramming it all through scanpy.

Addressing the above issues would strengthen the paper, the software, and the application of the software to users' papers. A fast, memory-efficient implementation of k-means is handy, but if few users apply k-means to cluster single-cell data going forward, then the application note is more of a technical demonstration of issues involved in applying in-memory algorithms to HDF5-backed out-of-core data. This is not a trivial task, but it's not clear to me whether the audience for it is the same as for an application note.

Reviewer #3: The authors developed an R package implementing the min-batch k-means algorithm

in a memory-efficient way, enabled application to on-disk data and integrated it

with widely used Bioconductor data containers in scRNA-seq analysis. The paper

itself is particularly well written, with methods, results and the specific

contributions of the work to the field presented clearly.

Their implementation of mbkmeans in an efficient Bioconductor package makes

great use of existing computational infrastructure and with full integration in

the Bioconductor ecosystem, mbkmeans is immediately useful and useable for

people doing single-cell data analyses in R. The example of mbkmeans to provide

an initial, fast clustering of cells before normalisation and further downstream

analysis steps is a very nice application.

Overall, we have a small number of comments, mainly with some

queries/suggestions where the authors may be able to provide a little more

clarity to ease understanding for the reader.

Of course, the authors quite deliberately avoid the lingering question for the

reader: "should I use k-means to cluster my single-cell data?" Fair enough for

answering that question to fall outside the scope of this paper; nevertheless it

would be very useful to know when k-means (esp as implemented in mbkmeans) is a

preferable choice for single-cell RNA-seq clustering over popular current

graph-based clustering methods like Louvain or Leiden.

1. Major comments on simulations

1.1 mbkmeans is fast and memory-efficient

It is clear that the original mini-batch kmeans algorithm is more memory

efficient compare to the traditional k-means. From the results in Fig1, it is

not so clear if the improvement in speed and memory consumption is achieved by

the algorithm itself or the implementation of mbkmeans. So it would be helpful

(say in a supp figure) to compare mbkmeans to the other R package that

implements the same algorithm, for example ClusterR, instead of traditional

k-means. It would be preferable to show results from the simulation experiment

in this section to speed and memory usage comparison with ClusterR.

1.2 mbkmeans is accurate

- Intuitively, it seems that the ratio between the batch size and the sample

size, instead of the absolute value of each, would influence the performance of

the algorithm. So it would be nice to have a figure showing the change of ARI

and/or WCSS w.r.t the change of the ratio.

- Can the authors provide a practical guide in how to select the batch size

w.r.t different sample sizes? Or a suggestion on a lower bound for the ratio

where a convergence in performance can be expected? Is the lower bound of 500

suggested in the text completely widely applicable, or are there any further

considerations for users to optimise this parameter in practice?

- This section seems to be evaluating the accuracy of the mini-batch algorithm,

does it mean any software implementing this algorithm reaches the same

accuracy? Or mbkmeans improved the accuracy compared to ClusterR?

- Does mini-batch k-means guarantee convergence?

- Are clustering results from different runs on the same data identical?

1.3 Impact of batch size on speed and memory-usage

Since the highlight of mbkmeans is its memory effectiveness, can the authors

provide an analytical expression (or heuristic) for the order of memory usage as

a function of the number of clusters k, sample size N and batch size b?

Otherwise, a rough explanation of how the memory usage and running time is

likely to change according to the change in k, N and b would also be very

helpful for users to manage their expectations. Understandable if the best

possible information is the results obtained under various settings in practice.

* Vignette

Overall, the vignette is excellent and easily allows a user to get going with

mbkmeans. We were able to install the package and run the code in the vignette

without any problems. We have the following minor notes:

- Missing "library" calls means that the user encounters errors if literally

running the code shown in the vignette; we suggest the authors add library()

calls for scater, DelayedMatrixStats, TENxPBMCData

- it would be handy to be able to enter a vector to the 'clusters' argument to

compute k-means for various k in one line. Otherwise, a short example in the

vignette showing how to neatly run mbkmeans with a range of k values would be

helpful, since this is a likely usecase for many users.

- A walk-through of using on-disk storage would be very handy (or a pointer to

resources for the user where they can read up on SCE objects with disk-backed

data)

2. Minor comments

2.1 on notation tweaks (Line 57, page 3)

- Since Y is a "set" of observations, it should be Y = {y_1, ..., y_N};

- "into k ( \\leq N) " could best be k (< N) since it makes no sense if k = N;

- Please mention ||.|| denotes L2 norm.

2.2 Possible typo: in abstract line 3, "analyses"  "analysis".

Thanks for the excellent paper,

PuXue Qiao and Davis J. McCarthy

**Have all data underlying the figures and results presented in the manuscript been provided?**

Reviewer #1: Yes

Reviewer #2: Yes

Reviewer #3: Yes

PLOS authors have the option to publish the peer review history of their article (what does this mean?). If published, this will include your full peer review and any attached files.

Reviewer #1: **Yes: **Vladimir Kiselev

Reviewer #2: No

Reviewer #3: **Yes: **Davis J. McCarthy
---

## [Decision Letter · Decision Letter 1]

10 Dec 2020

Dear Dr. Risso,

We are pleased to inform you that your manuscript 'mbkmeans: fast clustering for single cell data using mini-batch k-means' has been provisionally accepted for publication in PLOS Computational Biology.

Best regards,

Dina Schneidman

Software Editor

PLOS Computational Biology

Reviewer's Responses to Questions

**Comments to the Authors:**

Reviewer #2: The authors have addressed all of my remarks, and have gone above and beyond any reasonable standard for doing so. The improvements to both the package documentation and the manuscript are substantial, and reveal a degree of generality I had not initially appreciated. I commend the authors' efforts. I have no qualms about recommending the manuscript in its revised state for publication.

Reviewer #3: We are thoroughly satisfied with the authors' revisions based on our and other reviewers' comments. We congratulate them on an excellent paper and software tool and recommend publication.

**Have all data underlying the figures and results presented in the manuscript been provided?**

Reviewer #2: Yes

Reviewer #3: Yes

PLOS authors have the option to publish the peer review history of their article (what does this mean?). If published, this will include your full peer review and any attached files.

Reviewer #2: **Yes: **Timothy J. Triche, Jr.

Reviewer #3: **Yes: **Davis J. McCarthy

---

## [Editor Report · Acceptance letter]

21 Jan 2021

PCOMPBIOL-D-20-00959R1 

mbkmeans: fast clustering for single cell data using mini-batch k-means

Dear Dr Risso,

I am pleased to inform you that your manuscript has been formally accepted for publication in PLOS Computational Biology. Your manuscript is now with our production department and you will be notified of the publication date in due course.

With kind regards,

Alice Ellingham
